# Low Use of Long-Acting Reversible Contraceptives in Tanzania: Evidence from the Tanzania Demographic and Health Survey

**DOI:** 10.3390/ijerph19074206

**Published:** 2022-04-01

**Authors:** Amani Idris Kikula, Candida Moshiro, Naku Makoko, Eunyoung Park, Andrea Barnabas Pembe

**Affiliations:** 1Department of Obstetrics and Gynecology, Muhimbili University of Health and Allied Sciences, Dar es Salaam 11102, Tanzania; amanikikula@gmail.com (A.I.K.); andreapembe@yahoo.co.uk (A.B.P.); 2Department of Epidemiology and Biostatistics, Muhimbili University of Health and Allied Sciences, Dar es Salaam 11102, Tanzania; cmoshiro@gmail.com; 3School of Public Health and Social Sciences, Muhimbili University of Health and Allied Sciences, Dar es Salaam 11102, Tanzania; nakumakoko3@gmail.com; 4Department of Obstetrics and Gynecology, Wonju College of Medicine, Yonsei University, Wonju 26426, Korea

**Keywords:** long-acting reversible contraceptive, low use, Tanzania, demographic and health survey

## Abstract

We aimed to determine the prevalence and factors associated with the use of long-acting reversible contraceptives (LARCs) among women of reproductive age in Tanzania. We analyzed the Tanzania Demographic and Health Survey (DHS) data from 2015 to 2016. The study included 8189 women aged 15–49 years. The relationship between various factors and LARC use was determined through various analyses. Among women with a partner/husband, 7.27% used LARCs, 21.07% were grand multiparous, and 20.56% did not desire more children. Women aged 36–49 years (adjusted odds ratio (AOR)-2.10, 95% confidence interval (CI): 1.11–3.96), who completed secondary education (AOR-1.64, 95% CI: 1.05–2.55), who did not desire more children (AOR-2.28, 95% CI: 1.53–3.41), with a partner with primary level education (AOR-2.02, 95% CI: 1.34–3.02), or living in richer households (AOR-1.60, 95% CI: 1.12–2.27) were more likely to use LARCs. Further, women with a partner who wanted more children were less likely to use LARCs (AOR-0.69, 95% CI: 0.54–0.90). Tanzania has a low LARC usage rate. Women’s age, wife and partner’s education status, couple’s desire for more children, and household wealth index influenced the use of LARCs, highlighting the need to reach more couples of lower socioeconomic status to improve LARC utilization.

## 1. Introduction

Modern contraceptives include all products and medical procedures that interfere with sexual reproduction [1]. The World Health Organization (WHO) categorizes intrauterine contraceptive devices (IUCDs) and implants as long-acting reversible contraceptives (LARCs), among other modern contraceptive methods. Compared with other contraceptive methods, LARCs are highly effective over a prolonged period and offer an immediate return to childbearing potential upon removal; moreover, they can be used by all reproductive age groups [2].

In addition to the superior efficacy of LARCs over short-term contraceptive methods [2], they are more cost-effective after only 1 year of use [3]; the universal coverage for all women who require and are eligible for LARCs would generate a USD 120 billion annual return for each dollar spent [4]. In Sweden and the United States of America, reports have demonstrated remarkable economic benefits of LARCs over short-term methods [5,6]. In 2017, almost 4 million women in Tanzania were reported to use modern contraceptives, thereby preventing more than 1 million unplanned pregnancies and more than a third of a million unsafe abortions and avoiding more than 3000 maternal deaths [4,7]. These benefits would have been more pronounced with the use of more effective LARCs, and more so if they had been provided in a single visit [8].

Globally, multiple reports have documented several factors related to the use of LARCs. Maternal age was determined to be a factor, with teenagers less likely to use contraceptives than older women [9,10,11]. Excluding a select population in the USA [12], a higher educational status has also been linked to greater use of LARCs [11,13,14]; empowering women via education, therefore, seems to be a stronger mode of improving the use of contraceptives [14,15]. Lastly, a couple’s desire for more children bidirectionally affected the use of LARCs, with some positive [16] and some negative correlations associated with its use [11,17].

Tanzania has a high fertility rate (4.7%) that dropped from 5.2% in 2015/2016, and almost one-quarter of women intending to limit or postpone pregnancy do not use a reliable contraceptive method. This is accompanied by an unacceptably high maternal mortality ratio (556 per 100,000 live births) [18]. Over the years, the government of Tanzania and partner organizations have invested in family planning, among other reproductive and child health initiatives. Still, only one-third of eligible women use modern contraceptives, and in terms of LARCs, this rate is lower [7,19].

A review that examined the reasons for the high unmet need for contraception in Tanzania reported limited accessibility (3%) as the main hindrance [19]. Thus, the objective of this study was to determine the prevalence of LARC utilization and the influence of social factors among women in Tanzania (with a partner or husband) using the Tanzania Demographic and Health Survey (DHS) dataset. We expect that the results of this study will help Tanzania, as well as other countries in a similar situation, to improve LARC use upon consideration of the relevant identified factors.

## 2. Materials and Methods

### 2.1. Participants and Procedures

This was a cross-sectional study that utilized the Tanzania DHS data generated between 2015 and 2016 after obtaining permission from the DHS. We explored women’s data, household data, and partner/husband data to obtain the sociodemographic, economic, and obstetric characteristics.

Sampling was performed in two stages to develop a better estimate for Tanzania. The initial stage involved identifying clusters (608 identified), followed by the selection of households. From these, 22 households were systematically selected as representatives from each cluster, including a total of 13,376 households.

All women aged 15–49 years who were present in the sampled household as residents or visitors one night before the interview were eligible for inclusion. Regarding men, one-third of the sampled households were selected for an interview, of which all males in the sampled household who were present one night before the interview, either as a visitor or resident, were eligible for interview. The questionnaires were adapted from the standard international DHS program questionnaires and modified to suit the Tanzanian context.

Detailed information on sampling procedure and design was previously reported [18].

### 2.2. Measures

The outcome variable was the use of LARCs at the time of the interview, recorded as a binary variable (use: 1, nonuse: 0). The use of LARCs was a composite variable for the concurrent use of implants or IUCDs.

Independent variables comprised maternal age (<20, 20–35, or >35 years); husband’s/partner’s age (17–24, 25–34, 35–44, or >44 years); woman’s occupational status (unemployed, professional, clerical, agricultural (self-employed), agricultural (employee), household and domestic, services, skilled manual, or unskilled manual); woman’s educational status (no education, incomplete primary, complete primary, incomplete secondary, complete secondary, and higher); wealth quintile of the household as an indicator of socioeconomic status (poorest, poorer, middle, richer, or richest); husband’s/partner’s educational status (no education, primary, secondary, higher, or unknown); parity (0, 1–4, or >4); desire for last child (wanted then, wanted later, or wanted no more); desire for more children (wants within 2 years, wants after 2 years, wants (unsure timing), undecided, wants no more, sterilized and declared infecund); and husband’s/partner’s desire for children (both want the same, husband wants more, husband wants fewer, or unknown). These independent variables were controlled for women with partners/husbands to allow for the inclusion of the partner’s characteristics during the analysis.

### 2.3. Data Analysis

Data were analyzed using Stata version 14 (StataCorp, College Station, TX, USA). Due to the survey design of the dataset, sampling weights were accommodated during analysis to achieve the required point estimates.

Descriptive statistics, including frequencies and proportions, were used to summarize the data. Chi-squared tests were used to compare the prevalence of LARCs with independent variables, while logistic regression analyses were performed to determine the independent factors associated with the use of LARCs. Variables with *p* < 0.2 were included in the multivariable logistic regression model. The measure of association was estimated using crude and adjusted odds ratios (ORs) at 95% confidence intervals. All statistical tests were two-sided, and a *p*-value < 0.05 was considered statistically significant.

## 3. Results

### 3.1. Prevalence, Sociodemographic, Household, and Obstetric Characteristics with the Utilization of Long-Acting Family Planning (LARC)

The total number of women with husbands/partners included in the analysis was 8189; the response rate for the household interviews was 98% (12,563 households; 97% (13,266) for women, and 92% (3514) for men). The mean age of the female study participants was 31.53 (SD = 8.71) years, while that of the males (partners/husbands) was 38.43 (SD = 10.10) years. Among the 8189 women analyzed, 7.27% used LARCs at the time of the survey. While we considered all women of reproductive age, only 5.83% were using LARCs at the time of the survey; 21.07% were grand multiparous, and 20.56% did not desire more children. Among all households, 37.15% were poor, while 19.66% and 13.35% of women and their partners/husbands had no formal education, respectively (Table 1 and Table 2).

### 3.2. Factors Associated with Utilization of Long-Acting Family Planning (LARC)

Table 3 shows the factors associated with the use of the LARCs. All variables with a *p*-value ≤ 0.2 in the bivariable model were fitted to the logistic regression model. In this model, association with the use of LARC among reproductive-aged women with a partner was considered statistically significant at a *p*-value < 0.05. Statistically significant variables in the multivariable model included women’s age, women’s education level, desire for more children by women, wealth status, husband’s/partner’s desire for children, and partner’s/husband’s educational status.

As women’s age increased, LARCs were 2.6 times more likely to be used (AOR: 2.66, CI: 1.39–5.07); similarly, women who had completed secondary education were 1.6 times more likely to have a favorable attitude toward LARCs than those with no formal education (AOR: 1.64, CI: 1.05–2.55). Furthermore, women who desired children after 2 years and those who did not want more children were 2.3 times more likely to choose LARCs than women who wanted to conceive within 2 years (AOR: 2.23, CI 1.51–3.42 and AOR: 2.42, CI: 1.51–3.86, respectively). The partners/wives of men with formal education (primary level education and at least secondary school level) were 2.3 times and 2.4 times more likely to choose LARCs than those without formal education (AOR: 2.29, CI: 1.8–3.81 and AOR: 2.40, CI: 1.37–4.21), respectively. Conversely, the partners/wives of men who wanted more children were less likely to use LARCs (AOR: 0.67, CI: 0.49–0.91). Additionally, women living in the wealthier class households were 1.4 times more likely to use LARCs than those in the poorest households (AOR: 1.37, CI: 1.92–2.06).

Other variables, including parity, husband’s/partner’s age, desire for the last child, and woman’s occupational status, had no statistically significant association with LARC use.

### 3.3. Current Contraceptive Methods

Among the 8189 women analyzed in the survey, 7.27% used LARCs. For the 92.73% that did not use LARC their distribution was as follows, not using contraceptives (28.5%), injectables (41.7%), pills (15.1%), male condoms (14.7%). A visit to a health facility within 12 months or knowledge of family planning methods did not differ significantly in selecting LARCs. However, LARCs were not significantly selected among women informed about family planning (Figure 1 and Table 2).

## 4. Discussion

This study aimed to determine the prevalence and factors associated with the use of long-acting reversible contraceptives (LARCs) among women of reproductive age in Tanzania.

In this study, although a low LARC usage was observed in Tanzania, one in every five women of reproductive age was either grand multiparous or did not desire another pregnancy. Women’s age and educational status, both the woman and husband’s/partner’s desire for more children, husband/partner’s educational status, and household wealth index were significantly associated with the use of LARCs. In contrast, parity, husband’s/partner’s age, desire for the last child, as well as woman’s occupational status, good knowledge of LARCs, and less frequent medical visits were not significantly associated with LARC use.

Our study revealed a LARC usage level of 7.27%, which is considerably lower than reported in most African nations, including Egypt and Tunisia. While these countries exhibited approximately six times higher LARC utilization than Tanzania, use was higher in Tanzania than in South Sudan and Ivory Coast [20,21]. In Latin America, similar usage levels occur, with only a few nations demonstrating a prevalence of >10% [22]. Developed nations—including Norway—demonstrate relatively better LARC usage; however, the USA reports conflicting levels of just above or below 10% [12,20,23]. The lower usage rate in Tanzania (compared with other nations), coupled with the high unmet need for contraception [7], demonstrates a need to improve LARC use for benefits, including national economic development [4], as well as reduced maternal morbidity and mortality [4,7].

Women aged 20–35 years were more than twice as likely to use LARCs than teenagers. This may be because most women marry at an early age (median age: 19.2 years in Tanzania) [18], typically conceiving immediately after marriage and thus ruling out the need for LARCs in their teenage years. A similar relationship was observed in other sub-Saharan nations [11], Nigeria [9], the USA [10,24], and Indonesia [13]. Interestingly, women in Zambia aged below 25 years did not use LARCs [25]. However, reproductive health problems of adolescent girls (10–19 years, UN) also determine healthy childbirth and quality of life in adulthood. Key factors that are known to increase the chances of unsafe abortion of female adolescents in developing countries, violence against women, sexually transmitted diseases including HIV/AIDS, harmful traditional practices, nutritional disorders, and accidents are the most critical issues to be resolved. The need to strengthen youth-friendly family planning clinics for this vulnerable age group is thus necessary to extend the benefits of highly effective anticontraceptive tools [26]. Educational status was also found to influence LARC usage.

Women who completed secondary education were almost twice as likely to use LARCs than those with no formal education. This is due to the increased access to knowledge in the higher grades, which possibly created a demand to utilize healthcare services, including LARC. This advantage demonstrates the need for more effective and long-term contraceptive options even in the less formally educated women. These findings are similar to observations in Malawi, Zimbabwe [11], Indonesia [13], and most other low and middle countries [27]. While education level impacted the overall use of contraceptives in Ghana, Zambia, and Madagascar, it did not impact the choice of the method used [28]. In contrast, education level had no statistically significant association with the use of LARCs in the USA [12]. Education might improve the use of contraceptives in Tanzania—particularly in teenaged girls, as previously reported [14]—and is a key measure toward woman empowerment [15,29].

Women in households with a better wealth index were more likely to use LARCs than those in poorer households. This may be due to the accessibility of wealthier household members to quality (private) health care facilities that ensure stable availability of LARCs, evident in both Rwanda [11] and Latin American nations [22]. This illustrates the need to adapt family planning services to suit communities in the lower wealth index, allowing them to utilize these resources more effectively. Similarly, adaptable programs successfully improved the uptake of LARCs within Malawi and Zimbabwe [11]. However, the use of LARCs was not influenced by the wealth index in Indonesia [13]; however, Nigeria demonstrated opposite results, with a greater uptake in lower wealth index communities than their counterparts [30]. This is possibly due to the greater emphasis on promoting LARC use in rural communities, where extensive campaigns have taken place, than in urban areas, where most individuals pay for services that may have otherwise hindered their use.

Many wealthy people are relatively educated, live in urban areas, and have easier access to media and information, leading to a better understanding of the available methods and, thus, the use of LARCs. In contrast, a systematic review conducted in France and USA (Seattle) showed that women in difficult financial situations were positive toward IUCD use [31,32]. Together, these studies support the notion that both wealthy and poor women in the developed and developing world use LARCs. From this point, policymakers and health planners would be of paramount importance to focus on feasible strategies to effectively provide economic support and information in underdeveloped countries, empower women in economic activity, and provide subsidies for LARC in developed and developing countries.

Women who did not want more children were more than twice as likely to use LARCs than those who did want more children. Similarly, the wives/partners of men who wanted more children were less likely to use LARCs than those who did not want more children. This illustrates the need to adapt family planning clinics to be ‘male friendly’ since the women’s desire did not solely influence LARC use. A similar trend was observed in other sub-Saharan countries [11,17]; however, in Uganda, the desire for more children did not influence the use of LARCs [33]. In Nepal, a couple’s desire for more children promoted the use of LARCs, possibly due to the thought of an immediate return to fertility after cessation of using the contraceptive method [16]. Another reason may be a lack of awareness of the husband/partner. In studies conducted in Ethiopia’s Tigray and Oromia regions, lack of support from male partners and lack of discussion among partners was an obstacle to using LARC [34,35]. Married women with partners who did not allow the use of LARC, partners’ poor knowledge of the LARC and negative attitudes toward the LARC were negatively associated with LARC use. Male partners may not understand that LARC has long-term effectiveness and reversible benefits. Additionally, some may have misconceptions that do not support LARC utilization.

In this study, information and knowledge of methods on family planning did not affect Tanzanian women’s decision to use LARC. Unlike in previous similar studies, health care professionals, family planning providers’ advice, and discussions with health care providers about long-acting and permanent contraceptive methods had a significant impact on LARC utilization [36,37]. Providers have a responsibility to clearly communicate and support their clients to choose the method that best fits their personal circumstances. It would be good to provide in-service training for providers on how to support their clients in explaining the effectiveness of LARCs during their counseling sessions.

Husbands/partners who had at least a primary level of education were twice as likely to have a wife/partner that used LARCs than those with no formal education. This may be due to an improved understanding of its benefits, even at the basic level of formal education, as well as consideration of the impact of the male partner on reproductive health decisions in Tanzanian families. Similar findings were found in Nepal by Rajan et al. [16]; however, the husband’s/partner’s educational status did not influence LARC usage in Mozambique [17].

### Strength, Limitation and Recommend of the Study

This study tried to determine LARC utilization in the maximum possible number of individuals. The inclusion of partner/husband factors and household characteristics proved helpful; moreover, considering the structured nature of the sampling method, these results are valid and representative of the entire Tanzania.

Despite the observed results, this study had some limitations. One major limitation was the inability of the DHS data to capture provider factors, such as skills in family planning, counseling, and making available the required contraceptive method. These factors might influence the user’s experience, affecting the future use or nonuse of LARCs.

We also considered a specific group of women (matched women who had a partner/husband) for the analysis. This may have underestimated the prevalence of LARC use and effect measure of other independent variables since we excluded single women (introduction of Neyman bias).

These results should also be interpreted with caution, considering that there have been improvements in health services (from the time these data were collected to the writing of this manuscript) in Tanzania. This includes the number of facilities, quality of health services, and their providers. Nevertheless, these results are valid, considering the representative nature of the DHS data set and the standardized methodology; by matching women with their partners, partner characteristics for advanced statistical measures could be included. However, there is no new available DHS data set for such analysis; thus, these results will form a basis to focus on interventions and compare future DHS results.

Thus, we recommend future studies to explore healthcare provider factors that could affect the uptake of LARCs. Notably, more case studies are warranted to analyze the opportunities to maximize creation of demand for using LARC in all social classes in Tanzania.

## 5. Conclusions

Woman’s age, women and their husband’s/partner’s educational status, couple’s desire to have more children, and household wealth index influenced the use of LARCs. In contrast, parity, husband’s/partner’s age, desire for the last child, and woman’s occupational status, good knowledge of LARCs, less frequent medical visits were inversely associated with the use of LARCs. Tanzanian society would benefit from improved LARC use in several ways. Strategies to improve LARC adoption in Tanzania should therefore consider the social and economic characteristics of couples. The involvement of men in family planning education, counseling, and decision making regarding the benefits of LARC may ameliorate the partner’s negative impact on LARC use. From an economic point of view, Tanzania has a large gap between urban and rural areas. Therefore, it is possible to implement a strategy that maximizes the role of media in spreading awareness. Reproductive health in women might improve by reducing morbidity and mortality, which are associated with poor child spacing.

Tanzania’s high number of teenage pregnancies leads to interruptions in educational opportunities and deprivation of prospects to strengthen individuals’ socioeconomic capabilities while simultaneously accelerating poverty due to the continuing burden of childbearing and childrearing [18]. Therefore, youth-friendly services for teenage females, including overall sexual and reproductive health and providing an appropriate medical service infrastructure, improving the perception of healthcare, proper education, and employment, can ultimately improve women’s economic productivity, promote healthy childbirth, and boost the country’s competitiveness.

## Figures and Tables

**Figure 1 ijerph-19-04206-f001:**
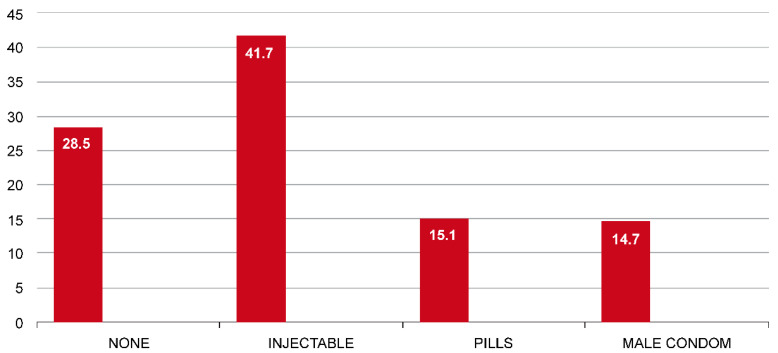
Current contraceptive methods used (in %) by Tanzania women if not using LARCs, 2015–2016.

**Table 1 ijerph-19-04206-t001:** Sociodemographic characteristics of participants related to long-acting family planning utilization (N = 8189).

Variable	Total	Using LARCs	Not Using LARCs	*p*-Value
**Woman’s age (years)**	N	*n* (%)	*n* (%)	
15–19	619	17 (2.75)	602 (97.25)	
20–35	4788	422 (8.81)	4366 (91.19)	<0.0001
>35	2782	156 (5.61)	2626 (94.39)	
**Husbands/Partners age (years)**				
17–24	624	36 (5.769)	588 (94.23)	
25–34	2590	226 (8.726)	2364 (91.27)	<0.0001
35–44	2629	212 (8.064)	2417 (91.94)	
>44	2345	121 (5.16)	2224 (94.84)	
**Woman’s occupation status**				
Not working	1388	77 (5.55)	1311 (94.45)	
Professional	281	21 (7.47)	260 (92.56)	
Clerical	34	1 (2.94)	33 (97.06)	
Agricultural—Self-employed	4101	300 (7.32)	3801 (92.68)	
Agricultural—Employee	237	17 (7.17)	220 (92.83)	0.1676
Household and domestic	263	20 (7.61)	243 (92.4)	
Services	285	27 (9.47)	258 (90.53)	
Skilled manual	337	28 (8.31)	309 (91.69)	
Unskilled manual	1263	104 (8.23)	1159 (91.77)	
**Woman’s education level**				
No education	1610	80 (4.97)	1530 (95.03)	
Incomplete primary	1026	73 (7.12)	953 (92.88)	
Complete primary	4022	323 (8.03)	3699 (91.97)	0.001
Incomplete secondary	643	39 (6.07)	604 (93.93)	
Complete secondary	810	75 (9.26)	735 (90.74)	
Higher	78	5 (6.41)	73 (93.59)	
**Husband’s/Partner’s education status**				
No education/don’t know	1093	36 (3.294)	1057 (96.71)	
Primary	5292	428 (8.088)	4864 (91.91)	<0.0001
Secondary and higher	1804	131 (7.262)	1673 (92.74)	
**Residence**				
Urban	2183	176 (8.06)	2007 (91.94)	
Rural	6006	419 (6.98)	5587 (93.02)	0.1321
**Wealth index**				
Poorest	1587	86 (5.42)	1501 (94.58)	
Poorer	1455	97 (6.67)	1358 (93.33)	
Middle	1586	126 (7.95)	1460 (92.06)	0.005
Richer	1835	164 (8.94)	1671 (91.06)	
Richest	1726	122 (7.07)	1604 (92.93)	

LARC—long-acting reversible contraceptive.

**Table 2 ijerph-19-04206-t002:** Obstetric characteristics of participants related to long-acting family planning utilization (N = 8189).

Variable	Total	Using LARCs	Not Using LARCs	*p*-Value
**Age at first birth (years)**				
≤13	94	89 (94.68)	5 (5.32)	
15–19	4492	4121 (91.74)	371 (8.26)	
20–24	2420	2244 (92.73)	176 (7.27)	
25–29	508	466 (92.46)	38 (7.54)	
≥30	108	105 (97.22)	3 (2.78)	0.141
**Parity**				
0	646	3 (.46)	646 (99.54)	
1–4	5149	428 (8.31)	4721 (91.69)	<0.0001
>4	2394	164 (6.85)	2230 (93.15)	
**Desire for the last child**				
Wanted then	3983	311 (7.81)	3672 (92.19)	
Wanted later	1519	220 (10.82)	1362 (89.66)	0.0112
Wanted no more	248	25 (10.08)	223 (89.92)	
**Desire for more children**				
Wants within 2 years	1862	58 (3.12)	1804 (96.89)	
Wants after 2 years	3507	328 (9.35)	3179 (90.65)	
Wants, unsure timing	81	3 (3.70)	78 (96.30)	<0.0001
Undecided	242	10 (4.13)	232 (95.87)	
Wants no more	2056	193 (9.39)	1863 (90.61)	
Sterilized/Declared infecund	438	3 (0.68)	435 (99.32)	
**Visited health facility in the last 12 m** **onths**				
Yes	5669	5246 (92.54)	423 (7.46)	
No	2518	2346 (93.17)	172 (6.83)	0.316
**Information on family planning provided**				
Yes	2184	200 (9.16)	1984 (90.84)	
No	3485	223 (6.40)	3262 (93.6)	0.0003
**Knowledge of family planning methods**				
Knows modern methods	8122	595 (7.33)	7527 (92.67)	
Don’t know	67	0	67 (100)	0.1115

LARC—long-acting reversible contraceptive.

**Table 3 ijerph-19-04206-t003:** Bivariate and multivariate analyses showing odds ratio for factors associated with the use of LARC (N = 8189).

Variable	N	COR (CI 95%)	AOR (CI 95%)
**Woman’s age (years)**			
15–19	619	1	1
20–35	4788	3.42 (2.09–5.61)	2.66 (1.39–5.07) *
36–49	2782	2.10 (1.25–3.54)	2.29 (1.08–4.87) *
**Husband/Partner age (years)**			
17–24	624	1	1
25–34	2590	1.56 (1.09–2.25)	0.99 (0.60–1.63)
35–44	2629	1.43 (0.97–2.11)	1.04 (0.59–1.81)
>44	2345	0.89 (0.60–1.32)	0.79 (0.41–1.51)
Parity			
0	646	1	1
1–4	5149	19.43 (6.27–60.17)	2.71 (0.37–20.07)
>4	1394	15.76 (5.07–49.03)	2.56 (0.33–19.64)
**Desire for more children (woman)**			
Wants within 2 years	1862	1	1
Wants after 2 years	3507	2.85 (2.18–3.72)	2.27 (1.51–3.42) *
Wants, unsure timing	81	0.45 (0.28–0.73)	-
Undecided	242	1.21 (0.75–1.96)	1.11 (0.46–2.69)
Wants no more	2056	3.21 (2.41–4.28)	2.42 (1.51–3.86) *
Sterilized/Declared infecund	438	0.47 (0.18–1.28)	-
**Desire for the last child**			
Wanted then	3983	1	1
Wanted later	1519	1.36 (1.10–1.68)	1.10 (0.85–1.42)
Wanted no more	248	1.32 (0.84–2.08)	1.16 (0.64–2.09)
**Woman’s occupation status**			
Unemployed	1388	1.38 (0.82–2.29)	0.93 (0.43–2.02)
Professional	281	0.52 (0.07–3.58)	1.03 (0.15–7.10)
Clerical	34	1	1
Agricultural—Self-employed	4101	1.34 (1.04–1.74)	1.21 (0.84–1.73)
Agricultural—employee	237	1.32 (0.74–2.33)	1.74 (0.83–3.66)
Household and domestic	263	1.40 (0.88–2.24)	1.25 (0.65–2.38)
Services	285	1.78 (1.15–2.77)	1.26 (0.69–2.29)
Skilled manual	337	1.54 (0.97–2.45)	1.16 (0.65–2.08)
Unskilled manual	1263	1.53 (1.51–2.03)	1.40 (0.98–2.01)
**Wealth index**			
Poorest	1587	1	1
Poorer	1455	1.25 (0.91–1.72)	1.28 (0.88–1.86)
Middle	1586	1.51 (1.08–2.10)	1.16 (0.75–1.78)
Richer	1835	1.71 (1.28–2.28)	1.37 (1.92–2.06) *
Richest	1726	1.33 (0.98–1.80)	0.93 (0.58–1.50)
**Husband/Partner desire for children**			
Both want the same	2966	1	1
Husband wants more	2138	0.69 (0.56–0.86)	0.67 (0.49–0.91) *
Husband wants fewer	444	1.26 (0.89–1.79)	1.35 (0.89–2.05)
Don’t know	2390	0.54 (0.44–0.68)	1.35 (0.89–2.05)
**Partner/Husband education status**			
No education	1093	1	1
Primary	5292	2.58 (1.82–3.67)	2.29 (1.8–3.81) *
Secondary and higher	1804	2.30 (1.55–3.41)	2.40 (1.37–4.21) *
**Woman’s education level**			
No education	1610	1	1
Incomplete primary	1026	1.33 (0.96–1.84)	1.36 (0.89–2.08)
Complete primary	4022	1.59 (1.23–2.07)	1.37 (0.99–1.88)
Incomplete secondary	643	0.76 (0.53–1.09)	1.05 (0.65–1.70)
Complete secondary	810	1.18 (0.85–1.65)	1.64 (1.05–2.55) *
Higher	78	1.09 (0.49–2.44)	1.32 (0.43–4.15)
**Residence**			
Urban	2183	1	
Rural	6006	0.86 (0.69–1.05)	-
**Knowledge of family planning methods**			
Knows modern methods	8122	0.63 (0.06–0.07)	
Don’t know	67	1	-
**Information on family planning provided**			
Yes	2184	1.47 (1.19–1.82)	1.17 (0.93–1.47)
No	3485	1	1

LARC—long-active reversible contraceptive; COR—crude odds ratio; CI—confidence interval; AOR—adjusted odds ratio; * statistically significant.

## Data Availability

Restrictions apply to the availability of data. Data were obtained from the DHS and are available from https://www.dhsprogram.com/data/dataset_admin (accessed on 28 July 2020) with the permission of the DHS.

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
