# Peer review of "Low Use of Long-Acting Reversible Contraceptives in Tanzania: Evidence from the Tanzania Demographic and Health Survey"

_ijerph, 2022, doi:10.3390/ijerph19074206_

Round 1
Reviewer 1 Report
The research problem is very interesting. I have a few doubts: 1. concerns the selection criteria for the study group (80-84): visitor one night before the interview were eligible for inclusion. How should this be understood? In my opinion, couples with a certain length of relationship should be studied. 2. The authors did not take into account cultural conditions. It would be necessary to show their specificity in Tanzania compared to other countries, e.g. Tunisia. 3. In my opinion, the variables studied do not contribute much to the understanding of the conditions of use The research problem is very interesting. I have a few doubts: 1. concerns the selection criteria for the study group (80-84): visitor one night before the interview were eligible for inclusion. How should this be understood? In my opinion, couples with a certain length of relationship should be studied. 2. The authors did not take into account cultural conditions. It would be necessary to show their specificity in Tanzania compared to other countries, e.g. Tunisia. 3. In my opinion, the variables studied do not contribute much to the understanding of the conditions of use LARCs. Relational variables, e.g. relationship satisfaction and other psychological variables, were missing.
Author Response
Thanks so much for your constructive suggestions, please see the attachment.

Reviewer 2 Report
The article entitled "Low use of long-acting reversible contraceptives in Tanzania: Evidence from the Tanzania Demographic and Health Survey" concerns the prevalence of LARC utilization and the influence of social factors among women in Tanzania. This is an important issue from a public health perspective.
However, I have some important doubts and comments related to them, which I describe below:
- In lines 56-59 the authors write: "Tanzania has a high fertility rate (5.2%), and almost one-quarter of women intending to limit or postpone do not use a reliable contraceptive method. This is 57 accompanied by an unacceptably high maternal mortality ratio (556 per 100,000 live 58 births) [18]. - However, the authors rely on the publication [18], which covers the period 2015-2016. Has the fertility rate in Tanzania changed in the years 2021-2022? if not - it should be explained, for example, that since 2015 the fertility rate has not changed. However, if a change has occurred, authors should also state it and explain it.
- 2. In lines 72-73 authors provide information about the current study: "This was a cross-sectional study that utilized the Tanzania DHS data generated between 2015 and 2016 after obtaining permission from the DHS - my main concern is why from this period? Although the economic situation of Tanzania is still weak and Tanzania is one of the poorest countries in the world, 6-7 years have passed from 2015-2016 to the present day. It is quite a long period in which changes could take place, both on the economic and social levels. Changes that, in turn, are important for the decisions currently being made about contraceptive methods. I do not deny the use of data from 2015-16, but the authors should explain why they decided to analyze the results from this period. Why did they not use up-to-date data? If only because such data has not (yet) been made available, the more the reasons why the authors undertake the analysis should be explained to the reader. The results based on data from several years ago should also be commented on in the context of the current socio-economic situation in Tanzania.
-
Line 143 - the abbreviations used in Table 3 are not explained: “LARC, long-active reversible contraceptive; COR, CI, confidence interval; AOR ". Please give the meaning of COR and AOR.
- Lines 146-147: "All variables with a p-value ≤0.2 in the bivariable model were fitted to the multivariable model" - the authors did not mark the variables for which p≤ 0.2. So it's hard to see which variables are mean? All? If not, they should also be included in the table, so that the reader does not have to check, e.g. in the Measures section or Tables 1 and 2, whether the variables in the table coincide with those included in the survey.
- Lines 149-152: "Variables included in the multivariable model were women's age, woman's education level, desire for more children of woman, wealth status, Husband's / Partner's desire for children, Partner's / Husband's educational status" - This description is incorrect . In the logistic regression model - according to the previous explanation (lines 146-147), there were "all variable with ap <0.2. The variables that the authors list in lines 149-152 were those that turned out to be statistically significant and therefore significant for the utilization of LARCs.
- Pie charts are not a good choice for scientific papers, thus, I recommend removing Fig 1. Information about the percentage distribution in the article is sufficient.
- Line 181: "Figure 2. Current contraceptive methods used by f Not using LARCs of Tanzania women, 2015-2016" - there is no information that the given values are percentages.
- Lines 220-221 - Incorrect font used for the word "education". Double-space should also be removed from the sentence.
- Line 315 - The authors probably mean "childbearing and childrearing" instead of "childbearing and childbearing".
Author Response
Thanks so much for your constructive comments, please see the attachment.

Reviewer 3 Report
This is a cross-sectional study discussing about the prevalence of LARC utilization and the influence of social factors among women in Tanzania. This study used a population-based, Tanzania Demographic and Health Survey dataset. I think the topic is important and contributive to the public health with an empirical approach quite valuable for public policy professionals.
Major concerns:
- Use of logistic regression is also inappropriate when using cross-sectional data with a common outcome. Any disorder with a prevalence of more than 10 percent will yield biased estimates of the relative risk when using the odds ratio. The authors should re-analyze with Proc Genmod with an appropriate model such as the negative binomial or exponential models.
- Please use power analysis to statement adequate sample size in this study.
- Lastly, the authors only briefly discuss limitations, acknowledging that the main limitation is the cross-sectional design. They should elaborate on how the use of this design is subject to Incidence-Prevalence bias, also known as Neyman bias, and how that might influence their findings.
- Some references are outdated and should be updated accordingly.
Author Response

(The authors gave the same response as above.)

Round 2
Reviewer 3 Report
No further comments. Thanks for your efforts on revision.